# Single-cell analysis reveals new evolutionary complexity in uveal melanoma

Michael A. Durante [1,2,3], Daniel A. Rodriguez [1,2,3], Stefan Kurtenbach[1,2,3], Jeffim N. Kuznetsov[1,2,3], Margaret I. Sanchez[1,2,3], Christina L. Decatur [1,2,3], Helen Snyder[4], Lynn G. Feun[2], Alan S. Livingstone[2,5] & J. William Harbour [1,2,3]*

Uveal melanoma (UM) is a highly metastatic cancer that, in contrast to cutaneous melanoma, is largely unresponsive to checkpoint immunotherapy. Here, we interrogate the tumor microenvironment at single-cell resolution using scRNA-seq of 59,915 tumor and non-neoplastic cells from 8 primary and 3 metastatic samples. Tumor cells reveal novel subclonal genomic complexity and transcriptional states. Tumor-infiltrating immune cells comprise a previously unrecognized diversity of cell types, including $CD8^+$ T cells predominantly expressing the checkpoint marker LAG3, rather than PD1 or CTLA4. V(D)J analysis shows clonally expanded T cells, indicating that they are capable of mounting an immune response. An indolent liver metastasis from a class 1B UM is infiltrated with clonally expanded plasma cells, indicative of antibody-mediated immunity. This complex ecosystem of tumor and immune cells provides new insights into UM biology, and LAG3 is identified as a potential candidate for immune checkpoint blockade in patients with high risk UM.

[1] Bascom Palmer Eye Institute, University of Miami Miller School of Medicine, Miami, FL, USA. [2] Sylvester Comprehensive Cancer Center, University of Miami Miller School of Medicine, Miami, FL, USA. [3] Interdisciplinary Stem Cell Institute, University of Miami Miller School of Medicine, Miami, FL, USA. [4] Cell IDx Inc., San Diego, CA, USA. [5] Department of Surgery, University of Miami Miller School of Medicine, Miami, FL, USA. *email: harbour@miami.edu

Uveal melanoma (UM) is a highly metastatic cancer that, in contrast to cutaneous melanoma, is largely unresponsive to checkpoint immunotherapy[1–3]. Here, we interrogate the tumor microenvironment (TME) at single-cell resolution using scRNA-seq of 59,915 tumor and non-neoplastic cells from eight primary and three metastatic samples. Analysis of tumor cells confirms the global genomic landscape established from bulk analysis and reveals newly described subclonal genomic complexity and transcriptional states consistent with phenotypic plasticity[4].

UM is notable for its well-characterized genomic landscape, high metastatic death rate, and resistance to therapy, including immune checkpoint inhibitors[5]. Prognostically distinct molecular subtypes have been identified based on gene expression profile (GEP), progression mutations, and chromosome copy number variations (CNVs)[6–10]. UMs with the class 1 GEP typically harbor mutations in *EIF1AX* (class 1A, low metastatic risk), *SF3B1*, or other splicing factors (class 1B, intermediate metastatic risk), and exhibit chromosomal gains of 6p and 8q. Those with class 2 GEP (high metastatic risk) are associated with inactivating mutations in *BAP1*, loss of chromosome 1p, 3, 6q and 8p, and gain of 8q. Based on computational inference from bulk sequencing data, UM appears to undergo an early punctuated evolutionary burst in which a full set of canonical aberrations arises specifying a particular subtype, after which further aberrations accrue as neutral passenger events[10].

## Results

**Single-cell RNA sequencing analysis.** To probe the TME at single-cell resolution, we performed droplet-based single-cell RNA sequencing (scRNA-seq) on 59,915 single cells from eight primary and three metastatic tumors, representing all GEP prognostic subtypes and BSE mutation categories (Fig. 1a, Supplementary Figs. 1, 2, and Supplementary Tables 1, 2). Dimensional reduction analysis using t-distributed stochastic neighbor embedding (t-SNE) reveals a diversity of tumor and non-neoplastic cell types (Fig. 1b and Supplementary Data 1). As expected, tumor cells cluster most strongly according to the GEP-based clinical prognostic classifier, with the primary division occurring between class 1 (*BAP1* wild-type) and class 2 (*BAP1* mutant) tumors (Fig. 1c). Individual tumors varied greatly in their composition, with cellular complexity increasing from primary class 1 to metastatic class 2 tumors (Fig. 1d). Interestingly, among the 12 genes comprising the validated GEP clinical prognostic test[11], five are expressed predominantly in tumor cells as expected (*EIF1B, HTR2B, ECM1, CDH1,* and *ROBO1*), but one is expressed predominantly in T cells (*SATB1*), and the remaining six are expressed in both tumor and immune cells (Supplementary Fig. 3 and Supplementary Data 1). These findings suggest that the accuracy of the GEP test may be due, at least in part, to its sampling of a transcriptional cross-section of this complex TME.

**Single-cell CNV analysis.** Next, we used CNVs as a means to probe the clonal structure of each tumor. The CNV content of individual cells was ascertained from scRNA-seq data using inferCNV, which was orthogonally validated against scDNA-seq (Supplementary Fig. 4a). Hidden Markov and Bayesian latent mixture modeling were performed to determine subclonal CNV events and remove low confidence CNV calls. This analysis reveals previously unappreciated complexity in both canonical and non-canonical CNVs (Fig. 2a–c and Supplementary Fig. 4b). While canonical CNVs dominate the chromosomal landscape as expected, there are multiple subclonal canonical and non-canonical CNVs across the samples. Surprisingly, class 1

tumors contain subclones of canonical class 2 CNVs (e.g. loss of 1p, 3, and 8p), and class 2 tumors contain subclones of canonical class 1 CNVs (e.g. gain of 6p and 6q). Further, we find evidence that canonical CNVs do not always occur in a single event but can arise from ongoing genomic evolution. For example, five cases (BSSR0022, UMM062, UMM063, UMM065, and UMM067L) show evidence for initial gain of 8q followed later by gain of 8p. In UMM065, loss of 3q in a 23% subclone is followed later by loss of 3p in a 6% subclone, resulting in LOH3. Despite harboring LOH3 cells, the GEP of this tumor is class 1, most likely because LOH3 is in a small subclone and a *BAP1* mutation has not occurred, consistent with the notion that the class 2 GEP requires LOH3 and mutation of *BAP1* on the other copy of chromosome 3 (ref. [12]). Previous studies showed that canonical genomic aberrations arise early in UM and give rise to one of three principal evolutionary trajectories associated with signature driver mutations—EIF1AX in class 1 A, SF3B1 and other splicing mutations in class 1B, and BAP1 in class 2 tumors[9,10], yet the single-cell resolution of our current findings reveal that these tumors continue to evolve with the development of heretofore unrecognized non-canonical CNV subclones that may contribute to tumor progression, as suggested by recent work[13].

**Transcriptional trajectory analysis.** In cutaneous melanoma, there is growing evidence that tumor cells undergo reversible switching between transcriptional states and that this plasticity drives metastasis and therapy resistance[4,14]. To elucidate transcriptional states across UM cells, we first analyzed scRNA-seq data using SCENIC[15] to identify potential co-expression modules and their associated *cis*-regulatory elements. The most over-represented motifs include those for oncoproteins MYC and JUN, as well as the bHLH-PAS hypoxia-associated transcription factor ARNT, all of which are enriched in cells of class 2 tumors (Fig. 3a). Then, we analyzed scRNA-seq data using Monocle 2 (ref. [16]), which reconstructs putative branching transcriptional trajectories to identify potential relationships across calculated states (Fig. 3b–d and Supplementary Figs. 5, 6). Pseudotime ordering of all tumor cells yields a total of 16 states organized into two main branches that self-assort according to GEP class 1 and class 2 (Fig. 3b, c). At the level of individual samples, cells from class 1 tumors are enriched in states 1-4,14-16 and those from class 2 tumors in states 5-13 (Fig. 3d), confirming the class 1/class 2 partition as a fundamental feature of the global molecular landscape of UM. We then analyzed the trajectories of each sample individually with Monocle 2 using branched expression analysis modeling (BEAM) and hierarchical clustering to identify genes enriched across states (Fig. 3e, Supplementary Fig. 6, and Supplementary Data 2). Transcriptional states are identified that are enriched for cells expressing HLA class I genes, consistent with previous work[17], as well as states associated with melanocyte differentiation (*PMEL*) and TNF-alpha/NF-kB signaling (*FOS, FOSB, JUN, JUNB, EGR1*). Within individual samples, these states are distributed across subclones and cell cycle phases (Supplementary Fig. 6), suggesting in vivo phenotypic plasticity that may be analogous to that described in melanoma cell lines[4].

**Immune cell and V(D)J immune repertoire analysis.** Adaptive plasticity of tumor cells among transcriptional states is driven by signals from the TME[4,14], which we investigated by generating a t-SNE plot of immune cells (Fig. 4a and Supplementary Fig. 7a) and performing hierarchical clustering of the average gene expression per cluster across tumor samples (Fig. 4b and Supplementary Fig. 7b–d). T cells are present in all samples, including CD8$^+$ cytotoxic T cells (mean, 0.2% class 1 versus 17% class 2 and metastatic) and CD8$^+$ T effector memory cells (mean,

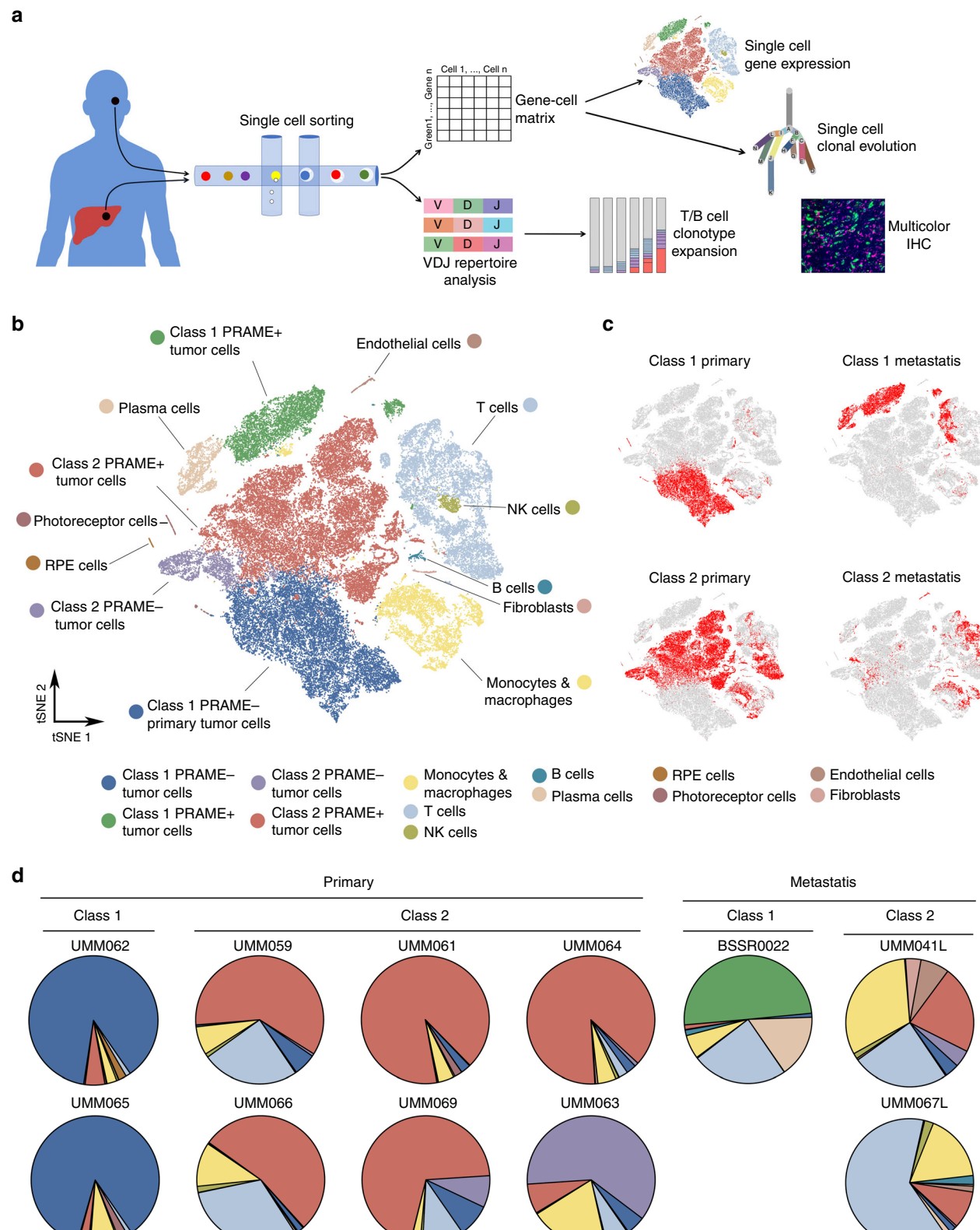

**Fig. 1 Aggregate analysis of 59,915 single cells from eight primary and three metastatic uveal melanomas. a** Summary of study design. **b** t-SNE plot of 59,915 single cells distributed by annotated unsupervised clustering. **c** t-SNE plot of 59,915 single cells highlighted by gene expression profile (GEP) class. **d** Pie charts of each of the eight primary and three metastatic tumors showing percentages of annotated cell types.

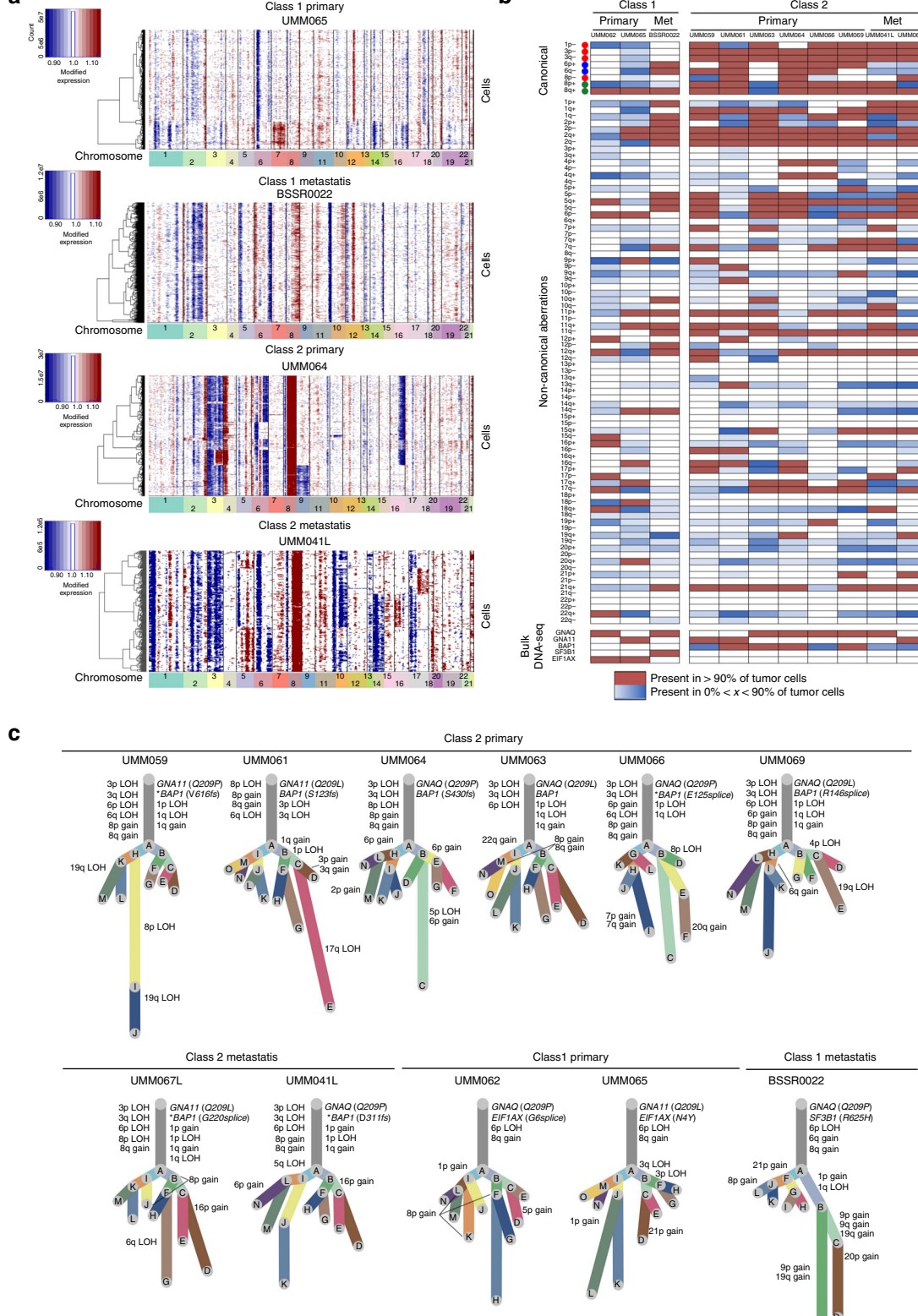

**Fig. 2 Single cell copy-number variation analysis of primary and metastatic uveal melanomas. a** Representative CNV heatmaps with hierarchical clustering from inferCNV analysis from each GEP class. **b** Summary plot of the CNV profiles from each of the 11 patients inferred from their scRNA-seq data. CNVs were annotated by the chromosome arm in which the CNV event calculated by inferCNV occurred. Canonical CNV events in UM are shown at the top as annotated (red, class 2; blue, class 1; green, class 1 and 2). Source data are provided as a Source Data file. **c** Clonality trees of each of the 11 patients separated by GEP class. The branches are scaled according to percentage of cells in the calculated subclone containing the corresponding CNVs. *indicates mutations that were found to occur in a subclone by bulk DNA sequencing and thus could not be assigned to a specific branch of the tree.

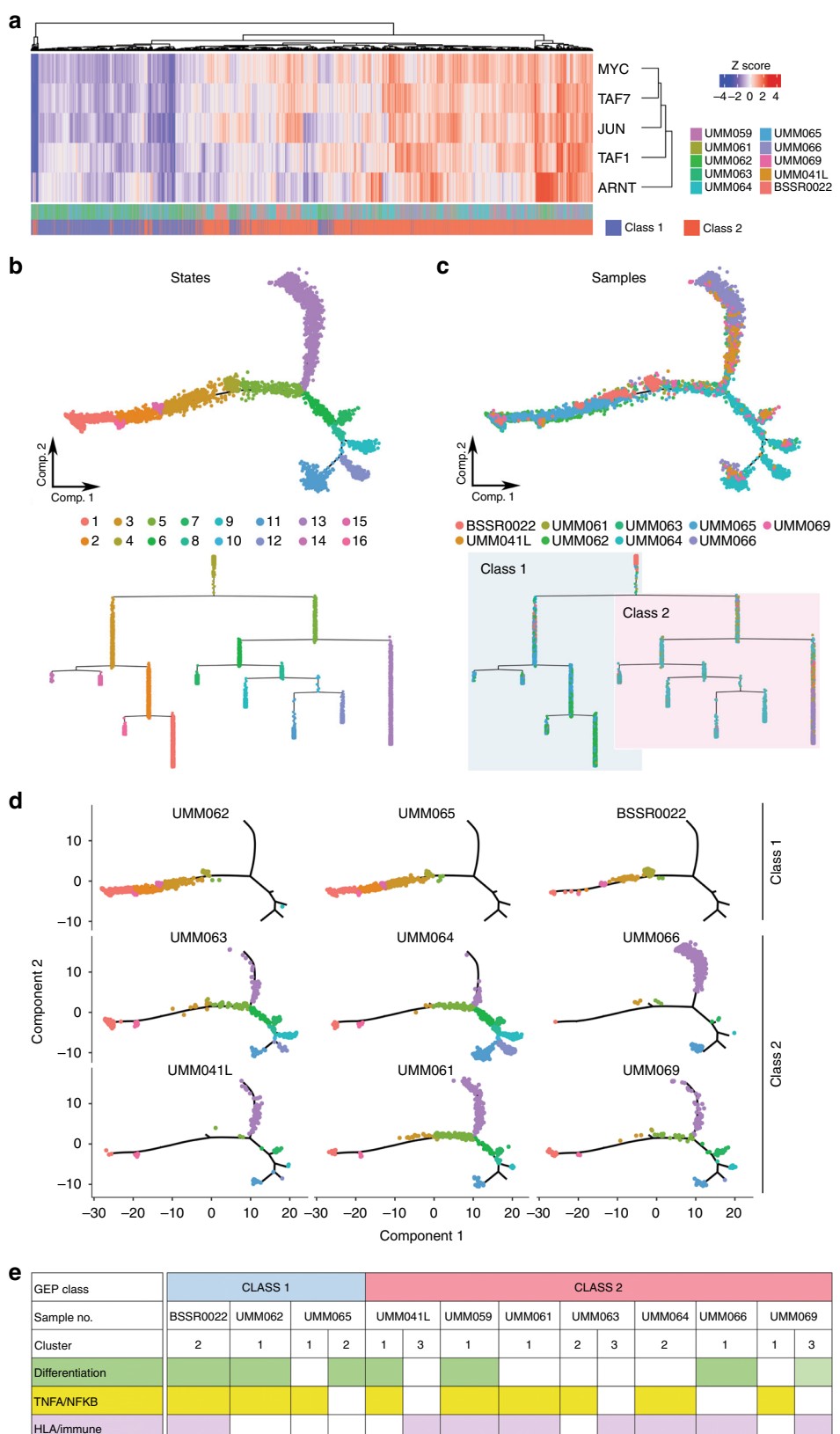

**Fig. 3 Trajectory analysis of uveal melanoma cells. a** Enriched transcription factor motifs determined using SCENIC analysis of 8,598 tumor cells and displayed as a heatmap of z-scored enrichment values. **b** Monocle 2 trajectory analysis of 7,947 uveal melanoma cells obtained by 5′ gene expression chemistry annotated by calculated states. **c** Trajectory analysis annotated by sample and overlaid with color by GEP class. **d** Combined Monocle 2 trajectory analysis displayed by each sample and annotated by calculated state. **e** Chart depicting pathways found to be significant by MSigDB analysis of each sample cluster determined by BEAM analysis followed by hierarchical clustering.

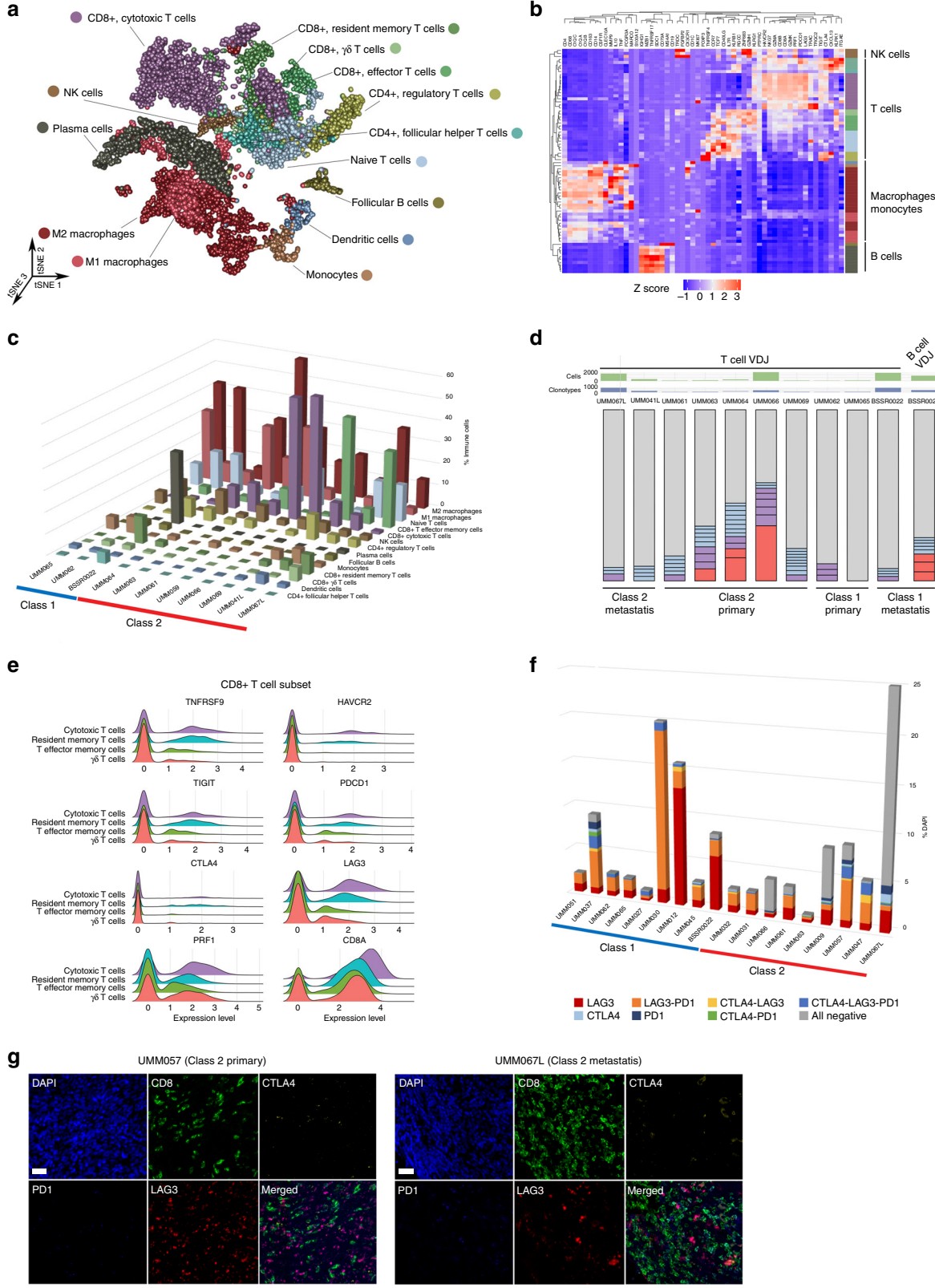

2% class 1 versus 12% class 2 and metastatic) (Fig. 4c and Supplementary Fig. 8a). Most T cells are CD8⁺, with smaller populations of CD4⁺ cells, including follicular helper cells, FOXP3⁺ regulatory cells, and naïve lymphocytes. V(D)J recombination analysis of T and B cell receptors from scRNA-seq data reveals clonally expanded T cells in only three samples (Fig. 4d), all class 2 primary tumors, but they are clonally expanded with exhausted T cells. CD8⁺ T cell expression of exhaustion-associated immune checkpoint molecules is strongest for *LAG3*, variable for *TIGIT*, and minimal for *PDCD1* (PD1), *CTLA4*, *HAVCR2* (TIM3), and *TNFRSF9* (Fig. 4e and Supplementary Fig. 7c, d). Protein expression of LAG3, CTLA4 and PD1 were orthogonally validated using multi-color IHC in 18 samples (Fig. 4f, g and Supplementary Fig. 8b). These findings, coupled with the low

**Fig. 4 Immune microenvironment of uveal melanomas with V(D)J recombination repertoire sequencing of B- and T- lymphocytes. a** t-SNE plot of 9441 single immune cells present in the TME. **b** Heatmap of averaged RNA expression of immune cell clusters. **c** Three-dimensional bar chart of immune cell subtypes as a percentage of immune cell population for each tumor. **d** Single-cell V(D)J recombination repertoire sequencing of T cells from 10 primary and metastatic UMs and B cells from an indolent class 1B metastasis. Red, clonotypes ≥4% T cell frequency; purple, clonotypes <4% and ≥2.5% T cell frequency; blue, clonotypes <2.5% and ≥1.5% T cell frequency; gray, all remaining clonotypes <1.5% T cell frequency. Source data are provided as a Source Data file. **e** Ridge plot of CD8+ T cell subset demonstrating strong expression of *LAG3*, moderate expression of *TIGIT*, and minimal expression of *PD1*, *CTLA4*, *TIM3*, and *TNFRSF9*. **f** Quantification of multi-color IHC for CD8, LAG3, PD1, CTLA4, and DAPI. 18 total samples were analyzed by IHC including 7 that were analyzed by scRNA-seq and an additional 11 samples. Metastatic samples include BSSR0022 and UMM067L. Other samples represent primary tumors. Quantitation of each sample was performed by whole-slide scanning of a single slide. Source data are provided as a Source Data file. **g** Representative multi-color IHC images of a primary and a metastatic class 2 UM stained for CD8, LAG3, PD1, CTLA4, and DAPI (scale bar, 50 μm).

expression of *PD-L1* and *PD-L2* in tumor cells (Supplementary Fig. 9a, b), may in part explain the ineffectiveness of CTLA4 and PD1 blockade in metastatic UM[1] and suggest a potential role for LAG3 in T cell exhaustion in UM. Similar to findings in other cancer types[18], *LAG3* is also expressed in some CD4+ T cells, FOXP3+ regulatory T cells, NK cells, and macrophage/monocytes (Supplementary Fig. 10). CD14+ monocytes/macrophages are present in all primary and metastatic samples, with CD68+ macrophages displaying a spectrum from M1- to M2-polarization (Fig. 4b, c and Supplementary Fig. 7b). Few NK cells are present, and they are distributed equally across tumor samples. B cells and plasma cells are rare in most samples. Remarkably, however, a provocative sample (BSSR0022) obtained from a solitary slow-growing liver metastasis arising 29 years after treatment of a primary class 1B tumor contains clonally expanded plasma cells, suggesting that the unusually protracted and indolent clinical behavior was facilitated by antibody-mediated immunity.

## Discussion

These findings reveal a complex ecosystem of tumor and immune cells and suggest that they co-evolve along trajectories associated with specific sets of genomic aberrations[10,19]. It is interesting to speculate that the long latency and low metastatic rate of class 1 UMs may be due, at least in part, to immune surveillance, which could result from neoantigens generated by *EIF1AX* and *SF3B1* mutations[20]. This possibility could be of clinical significance and warrants further investigation. By contrast, we hypothesize that the canonical genomic aberrations and increased overall aneuploidy in class 2 tumors create an immunosuppressive microenvironment that promotes metastasis through immune escape. Consistent with this possibility, recent work has linked aneuploidy to immune suppression and immunotherapy resistance through gene dosage effects caused by arm level CNVs like those seen in UM[21] and by activation of NF-kB via the cGAS-STING cytosolic DNA pathway[22]. It is interesting to speculate that one or both of these mechanisms explain the association between aneuploidy, metastasis and dysfunctional immune infiltrates in class 2 UMs.

Our scRNA-seq V(D)J analysis showing clonally expanded T cells and/or plasma cells in UM samples indicates that tumor infiltrating immune cells are capable of mounting a response, suggesting that low tumor mutation is not the only explanation for the poor response of UMs to checkpoint inhibitors. Indeed, our discovery of LAG3 as the dominant exhaustion marker in UM may explain, at least in part, the failure of previous checkpoint blockade targeting CTLA4 and PD1 (ref. [1]). LAG3 is the third immunoinhibitory receptor to be targeted in patients, demonstrates considerable synergy with PD1, is expressed not only on CD8+ T cells but also on NK cells and regulatory T cells, and has unique properties that could significantly expand the efficacy of checkpoint inhibitor therapy[18]. As such, LAG3 inhibitors are being evaluated in a large number of clinical trials in multiple cancer types[23].

## Methods

**Patients and sample collection.** Human tissue samples were obtained with patient informed consent and approval of the Institutional Review Board of the University of Miami. Immediately following surgical eye removal or liver resection, the tissue was dissected to isolate the tumor region for single-cell dissociation. Metastatic tumor tissue was intentionally sampled far from the tumor-liver interface to avoid contaminating normal liver tissue. Additional tissue samples were taken from the tumor for DNA and RNA profiling. These samples were subjected to DNA extraction using the Wizard Genomic DNA Purification kit (Promega, Madison, WI) and RNA extraction using the PicoPure RNA Isolation kit (Thermo Fisher Scientific).

**Tissue processing for single-cell suspension.** Tissue samples were placed immediately in gentleMACS C tubes (Miltenyi Biotec) containing 5 mL of DMEM or RPMI1640 Media with 10% FBS and 400 U/mL of collagenase IV for digestion[24]. The "Dissociation of soft tumors" protocol from the Miltenyi Tumor Dissociation Kit was used with a slight modification. Briefly, samples were processed using a gentleMACS dissociator (Miltenyi Biotec) using program "h_tumor_01" and incubating at 37 °C for 1 h. Samples were processed again using program "h_tumor_03" and passed through a 70 μm cell strainer (Miltenyi Biotec). After the initial incubation step cells were kept on ice for the remainder of the protocol. Wide Orifice 1 mL Pipet Tips (VWR) were used to prevent cell shearing. The cell suspension then underwent a Debris Removal Solution (Milteni Biotec) protocol, a density gradient method to remove dead cells and debris. The samples were resuspended in D-PBS containing 0.1% BSA and filtered into Falcon Tubes (12 × 75 mm) with Cell Strainer Cap (BD). An aliquot of the singe cell suspension was stained with the LIVE/DEAD™ Viability/Cytotoxicity Kit for mammalian cells (Invitrogen) to ensure viability was greater than 85%. Samples were processed within 3 h from surgical removal to loading on the Chromium (10X Genomics) instrument. We acknowledge that dissociation-associated artefacts may exist and developed an optimized protocol to efficiently process these UM samples consistently[25].

**Single-cell RNA sequencing.** Single-cell RNA sequencing was performed using the Chromium (10X Genomics) instrument. Single cell suspensions were counted using both the Cellometer K2 Fluorescent Viability Cell Counter (Nexcelom) and a haemocytometer and adjusted to 1000 cells/μl. UMM061, UMM062, UMM063, UMM064, UMM065, UMM066, UMM069, UMM067L, UMM041L, and BSSR0022 were run using the Chromium Single Cell 5' Library & Gel Bead Kit v2, Chromium Single Cell V(D)J Human T Cell Enrichment Kit, and Chromium Single Cell V(D)J Human B Cell Enrichment Kit, (10X Genomics). UMM059 was run using the Chromium Single Cell 3' Library & Gel Bead Kit v2 (10X Genomics) which is not compatible with the Chromium Single Cell V(D)J product. The manufacturer's protocol was used with a target capture of 10,000 cells for the 5' gene expression samples and a target capture of 5,000 cells for the 3' gene expression sample (UMM059). Each sample was processed on an independent Chromium Single Cell A Chip (10X Genomics) and subsequently run on a thermocycler (Eppendorf). 3' and 5' gene expression libraries were sequenced using the NextSeq 500 150-cycle high-output flow cells. B- and T- cell VDJ libraries were sequenced on a MiSeq instrument.

**Single-cell DNA sequencing.** Single-cell DNA sequencing was performed using the Chromium instrument. Single cell suspensions were counted using both the Cellometer K2 Fluorescent Viability Cell Counter and a haemocytometer and adjusted to 1,000 cells/μl. UMM069 and UMM041L were run using the Chromium Single Cell DNA Library & Gel Bead Kit (10X Genomics) with a target capture of 500 cells. The samples were processed on Chromium Single Cell C and D Chips (10X Genomics) according to the manufacturer's protocol and subsequently run on a thermocycler. Single-cell genomic DNA libraries were sequenced using the NextSeq 500 300-cycle high-output flow cells.

**Patient tumor RNA and DNA profiling.** Tumor RNA samples were subjected to gene expression profiling using the DecisionDx®-UM test (Castle Biosciences, Inc.), as previously described[11]. Tumor DNA mutations were determined using the DecisionDx-UMseq® targeted next-generation sequencing panel (Castle Biosciences, Inc.).

**Single-cell RNA sequencing analysis**. Raw base call (BCL) files were analyzed using CellRanger (version 2.1.1). The "mkfastq" command was used to generate FASTQ files and the "count" command was used to generate raw gene-barcode matrices aligned to the 10X Genomics GRCh38 Ensembl build 84 genome (version 1.2.0). The data from all 11 samples were combined in R (3.5.1, 3.5.2) using the Read10X() function from the Seurat package (2.3.4) and an aggregate Seurat object was generated[26,27]. Filtering was conducted by retaining cells that had unique molecular identifiers (UMIs) greater than 400, expressed 100 and 8000 genes inclusive, and had mitochondrial content less than 10 percent. No sample batch correction was performed. This resulted in a total of 59,915 cells. Data were normalized using the "LogNormalize" method and using a scale factor of 10,000. Using Seurat's Scale.Data () function and "vars.to.regress" option UMI's and percent mitochondrial content were used to regress out unwanted sources of variation. Cell cycle analysis was conducted using the CellCycleScoring() with a list of cell cycle markers, from Tirosh and colleagues[28]. The number of variably expressed genes were calculated using the following criteria: normalized expression between 0.125 and 3, and a quantile-normalized variance exceeding 0.5. To reduce dimensionality of this dataset, the resulting 1865 variably expressed genes were summarized by principle component analysis (PCA), and the first 20 principle components further summarized using t-distributed stochastic neighbor embedding (tSNE) dimensionality reduction[29]. The RunTSNE() wrapper function was used with the Barnes-Hut implementation of the 'Rtsne' package (0.15). Doublets were assessed using the DoubletFinder (2.0.2)algorithm[30] (Supplementary Fig. 1d) and few (<10%) doublets were observed outside of the macrophage/monocyte population. Clustering was conducted with the FindClusters() function using 20 PCA components and a resolution parameter set to 3. The original Louvain algorithm was utilized for modularity optimization[31]. The resulting 58 louvain clusters were visualized in a two-dimensional tSNE representation and were annotated to known biological cell types using canonical marker genes (Fig.1 and Supplementary Fig. 1). Tumor cells were identified using *MLANA*, *MITF*, and *DCT*. Tumor cells were further divided into subgroups by expression of *PRAME* and GEP genes (Supplementary Fig. 1). The following cell types were annotated using: T Cells (*CD3D*, *CD3E*, *CD8A*), B cells (*CD19*, *CD79A*, *MS4A1* [*CD20*]), Plasma cells (*IGHG1*, *MZB1*, *SDC1*, *CD79A*), Monocytes and macrophages (*CD68*, *CD163*, *CD14*), NK Cells (*FGFBP2*, *FCG3RA*, *CX3CR1*), Retinal pigment epithelium (*RPE65*), Photoreceptor cells (*RCVRN*), Fibroblasts (*FGF7*), and Endothelial cells (*PECAM1*, *VWF*). For the immune cell subset analysis, the SubsetData() function was used with "do. clean" set to TRUE and the previously identified cell types (T cell, NK cell, B cell, Plasma cell, monocyte and macrophage). This resulted in 16,740 immune cells that were normalized using the "LogNormalize" method with a scale factor of 10,000. The number of variably expressed genes were calculated using the following criteria: normalized expression between 0.125 and 3, and a quantile-normalized variance exceeding 0.5. The 4423 variably expressed genes were summarized by PCA, and the first 20 principle components further summarized using tSNE as described above. Clustering was conducted using 20 PCA components and a resolution parameter set to 10. The original Louvain algorithm was utilized for modularity optimization. The resulting 74 louvain clusters were used as input to the AverageExpression() function to generate average RNA expression data for each cluster. Hierarchical clustering was conducted on the RNA averaged clusters with immune cell genes aggregated from the literature[32,33] and visualized using a heatmap (Fig. 3, Supplementary Fig. 3)[34]. The cell types described above clustered similarly with hierarchical clustering with the corresponding immune cell genes. Immune cell subpopulations were identified using genes previously reported to identify the following populations: T regulatory cells (*FOXP3*, *TNFRSF4*, *IKZF2*, *IL2RA*), Follicular T cells (*CD200*, *GNG4*, *CHN1*, *IGFL2*, *ITM2A*, *CPM*, *NR3C1*), Naive T cells (*IL7R*), CD8+ T effector memory cells (*CD8A*, *ZNF683*), CD8+ resident memory cells (*KLRK1*, *ITGAE* [*CD103*]), Cytotoxic CD8+ T cells (*PRF1*, *GZMA*, *GZMK*, *NKG7* with varying levels of expression of the exhaustion markers *LAG3*, *PD1*, *CTLA4*, *TIGIT*, *HAVCR2* [*TIM3*], and *TNFRSF9* [*4-1BB*]), CD8+ gamma delta T cells (*TRDC*, *TRDG2*), Mitotic CD8+ T cells (*MKI67*, *STMN1*, *HMGB2*, *TUBB*, *TUBA1B*), Dendritic cells (CD1C and lack of expression of *C1QA*, *C1QB*, and *C1QC*), Monocytes (*S100A12*, *CLEC10A*), M2 macrophages (*CD163*, *C1QA*, *C1QB*, *C1QC*, *IL10*), M1 macrophages (*C1QA*, *C1QB*, *C1QC*, lack of M2 macrophage markers). Mitotic macrophages (*MKI67*). B cells, plasma cells, NK cells were identified as described above. These genes were annotated with data generated from the FindAllMarkers() Seurat differential expression analysis using the default two-sided non-parametric Wilcoxon rank sum test with Bonferroni correction using all genes in the dataset (Supplementary Data 1 and Supplementary Data 2) and a literature review[32,33]. To generate the ridge plots (Fig. 4) and dot plots (Supplementary Fig. 7), Seurat (3.0.0) was used on the CD8+ T cell subset. Seurat (3.0.0) was only used to generate plots with data analyzed with Seurat (2.3.4). The Update-SeuratObject(), RidgePlot(), and DotPlot() functions were utilized to generate the plots from the Seurat 2 dataset generated above. Additional packages used for data analysis include: ggplot2 (2.3.1,3.1.1), dplyr (0.7.8, 0.8.0.1), Rtsne (0.15), forcats (0.3.0, 0.40), bindrcpp (0.2.2), cowplot (0.9.3), Matrix (1.2-16), scales (1.0.0), jpeg (0.1-8), colorRamps (2.3), paletter (0.0.0.9000), cellranger (1.1.0), DDRTree (0.1.5), psycho (0.4.9), tidyverse (1.2.1), tibble (2.1.1), VGAM (1.1-1), irlba (2.3.3), stringr (1.4.0), Biobase (2.42.0), purrr (0.3.2), readr (1.3.1), ggpubr (0.2v).

**Single-cell CNV analysis**. Raw BCL files for the DNA sequencing data were processed using Cellranger DNA (version 1.0.0). The "mkfastq" command was used to generate FASTQ files and the "cnv" command was used to generate CNV data

aligned to the 10X Genomics GRCh37 build 87 genome (version 1.0.0). Results were visualized in the Loupe scDNA Browser (version 1.0.0). Cells filtered using subtree depth of 4 for UMM069 and subtree depth of 6 for UMM041L by removing diploid cells attributed to immune infiltrate and other non-tumor cell types.

**inferCNV and clonality analysis**. Raw gene expression data were extracted from the Seurat object as recommended in the "Using 10x data" section (inferCNV of the Trinity CTAT Project, https://github.com/broadinstitute/inferCNV). For each patient, normal reference cells were selected by expression of *CD3E* greater than 2 standard deviations above the mean expression and no expression of *PRAME* and *HTR2B*. Melanoma cells were identified from the annotated Louvain clusters as determined above. Quality control filtering was performed to select the highest quality cells by only including melanoma cells with greater than 3000 UMIs. For the inferCNV analysis the following parameters were used: "denoise", default hidden markov model (HMM) settings, and a value of 0.1 for "cutoff". To reduce the possibility of false-positive CNV calls the default bayesian latent mixture model was implemented to identify the posterior probabilities of alterations in each cell. Low-probability CNVs were filtered using the default value of "0.5" for the threshold. To determine the clonal CNV changes in each tumor the "subcluster" method was utilized on the CNVs generated by the HMM. GRCh38 cytoband information was used to convert each CNV to a p- or q- arm level change for simplification based on its location. Each CNV was annotated to be either a gain or a loss. After data conversion, subclones containing identical arm level CNVs were collapsed and trees were restructured to accurately represent subclonal CNV architecture. Mitochondrial CNVs were excluded from this analysis. For data visualization, the UPhyloplot2 (https://github.com/harbourlab/UPhyloplot2) plotting algorithm was developed to automate generation of intra-tumor evolutionary trees. The arm level CNV calls curated from the inferCNV HMM subcluster CNV predictions algorithm and the percentage of cells in each of the subclones were used as inputs. A scalable vector graphics (.svg) file visualizing the phylogenetic tree was generated for each sample. Arm length is proportional to percentage of cells plus a spacer (circle diameter + 5 pixels). Driver mutations and their mutant allele frequencies were determined using the DecisionDx-UMseq® targeted next-generation sequencing UM panel (Castle Biosciences, Inc.). Mutant allele frequencies of the *BAP1*, *SF3B1*, and *EIF1AX* were corrected for normal contamination by setting the mutant allele frequencies of *GNAQ* or *GNA11* to 50%. Mutations in UM driver genes were attributed to the CNV clones of like percentages and were marked when a specific clone could not be determined.

**SCENIC analysis**. The pySCENIC (0.9.9 + 2.gcaded79) algorithm was run on a normalized expression matrix of the 8,598 high-quality UM cells[15]. The GRNboost2 (arboreto 0.1.5) method was utilized for gene regulatory network reconstruction[35]. The cisTarget Human motif database v9 (https://resources. aertslab.org/cistarget/motif2tf/motifs-v9-nr.hgnc-m0.001-o0.0.tbl) of 24,453 motifs were used for enrichment of gene signatures and pruned for targets from this signature based on cis-regulatory cues with default settings. The "aucell" positional argument was utilized to find enrichment of regulons across single cells. The resulting matrix was z-scored using the standardize() function from the psycho (0.4.9) R package and the results were visualized using a heatmap with hierarchical clustering[34].

**Monocle 2 analysis**. UM-specific cells were identified from the annotated Louvain clusters as determined above and filtered for cells expressing any of the following immune cell markers (*IGHG1*, *CD3E*, *CD68*, *CD163*, *LYZ*, *MS4A1*, *CD79A*, *CD14*, *C1QA*). Only 5′ gene expression data were considered to prevent chemistry-related artefacts. This resulted in 7,947 high-quality UM cells to use for this analysis. Single-cell pseudotime trajectories were constructed with Monocle 2 (2.10.1)[16,36]. For the individual trajectory analyses, we utilized the normalized expression data from each sample. Genes for trajectory inference were selected using the dispersionTable() function to calculate a smooth function describing how variance in each gene's expression across cells varies according to the mean. Only genes with mean expression greater than or equal to 0.1 were used for the analysis. The reduceDimension() function was utilized with the DDRTree[16] reduction method and the following parameters modified: max_components = 3, and num_dim = 20. Results were visualized using the plot_cell_trajectory() and plot_complex_cell_trajectory() functions and annotated with cell cycle, subclones less than or equal to 20%, and calculated cell states. To identify genes that separate cells into the calculated states we used the BEAM() function to perform BEAM. Genes resulting from the BEAM analysis with a q-value less than or equal to 0.01 were separated with hierarchical clustering using the plot_multiple_branches_heatmap() function with num_clusters = 3 and "branches" set to the terminal branchpoints for each respective sample. Genes from each respective hierarchical cluster were input into the "Compute Overlaps" Gene Set Enrichment Analysis software (v6.4), which calculates a False Discovery Rate using the Benjamini Hochberg method to correct the hypergeometric p-value (http:// software.broadinstitute.org/gsea/msigdb/annotate.jsp)[37]. The Hallmark, C1, C2, C3, C4, C5, C6, and C7 MSigDB gene sets were used in this analysis[38]. For the combined trajectory analysis, we utilized the same parameters discussed above except the reduceDimension() function included the ncenter = 600 parameter.

**Deparaffinization, staining, and imaging**. Deparaffinization and rehydration of FFPE sections involved sequential incubation of the slides in xylene ($2 \times 5$ min), 100% ethanol ($2 \times 2$ min), 95% ethanol ($1 \times 2$ min), tap water ($2 \times 2$ min) followed by distilled water ($1 \times 2$ min). Antigen retrieval involved placing slides in a staining container of 10 mM citrate buffer, 0.05% Tween 20 pH 6, inside a pressure cooker and steaming under high pressure at approximately 110–120 degrees for 15 min. Slides were then cooled in the pressure cooker for 10 min before releasing the steam and placed in hot distilled water for 2 min prior to rinsing under running tap water for 5 min followed by wash buffer (PBS/0.2% Tween 20, pH 7.2) for 5 min. Excess liquid was removed from the slides and a barrier drawn around the tissue section using a hydrophobic pen. Sections were blocked by incubating with blocking buffer (PBS/3% normal rabbit serum/0.1% TritonX) for 20 min. Blocking buffer was aspirated and a cocktail of primary antibodies (UltraPlex detection system, Cell IDx) diluted with antibody diluent (PBS/1% BSA/0.2% Tween 20/15 mM) was added to the slide and incubated for 1 h at room temperature in a humidified chamber. Slides were then washed with wash buffer ($3 \times 5$ min) and a cocktail of detection antibodies (UltraPlex detection system, Cell IDx) diluted with antibody diluent (PBS/1% BSA/0.2% Tween 20) was added to the slide and incubated for 1 h as previously. As negative control, slides were incubated with secondary detection cocktail alone. Slides were then washed with wash buffer ($3 \times 5$ min) and rinsed with distilled water ($1 \times 2$ min). Slides were then mounted using Fluoroshield with DAPI (Immunobiosciences) and coverslips applied prior to scanning at 20X using the Leica Versa scanner. Analysis was performed on the Aperio ImageScope, (v12.4.2.5010), using Leica Quantitative Algorithm (v1).

**UltraPlex antibodies**. Each primary antibody was labeled with a specific peptide hapten tag (UltraPlex, Cell IDx) and combined at a final concentration of 5ug/ml of each antibody. The UltraPlex panel comprised CTLA-4-CH014 (clone CAL49), CD8-CH015 (clone EP334), PD-1-CH016 (clone EP239), and LAG3-CH021 (clone EP294). The detection cocktail comprised anti-CH014-CL550, anti-CH015-CL490, anti-CH016-CL750 and anti-CH021-CL650. All secondary antibodies were combined at a final concentration of 5 µg/ml for each antibody.

**Single-cell V(D)J analysis**. Raw BCL files for each B-cell and T-cell library were analyzed using CellRanger (version 2.2.0). The "mkfastq" command was used to generate FASTQ files and the "vdj" command was used to generate sequence annotations and VLOUPE visualization files aligned to the GRCh38 Ensembl build 87 genome in addition to a 10X-specific addendum to genes and a 10X-specific blacklisted transcript ID (2.0.0). Raw data from each sample from the "all_contig_annotations.csv" output were intersected with the T and B cells previously filtered using Seurat. Further filtering of the data was conducted by only including clonotypes that had "productive","high_confidence", and "is_cell" equal to true. Clonotypes were grouped by "raw_clonotype_id" for clonotype percentage determination. Additionally, the "chain", "v_gene", "d_gene","j_gene","c_gene", and "cdr3" sequences were collected for each clonotype and are available as Supplementary Table 2. Sequence level clonotype data were also investigated using the Loupe VDJ Browser (2.0.1). To visualize the distribution of each clonotype in tSNE space the Loupe Cell Browser (2.0.0) was used. A loupe file of Seurat filtered T/B cells was generated using Cellranger reanalyze and the VDJ analysis vloupe file was imported to show clonotypes present and confirm clonotype percentages.

**Statistics**. No statistical method was used to predetermine sample size. For each experiment, tumor tissue samples from a single patient were processed individually. Single cell suspensions for each sample were processed for scRNA-seq (10x Genomics) in an independent Chromium chip. For differential expression analysis in Seurat, the default two-sided non-parametric Wilcoxon rank sum test with Bonferroni correction using all genes in the dataset was utilized.

**Reporting summary**. Further information on research design is available in the Nature Research Reporting Summary linked to this article.

## Code availability

For visualization of the intra-tumor evolutionary trees, the UPhyloplot2 plotting algorithm was developed and is available at https://github.com/harbourlab/UPhyloplot2.

## Data availability

All sequencing data generated have been deposited in dbGaP under accession code phs001861.v1.p1. Processed sequencing data have been deposited in GEO under accession code GSE139829. The cisTarget Human motif database v9 used as part of the SCENIC analysis can be accessed at https://resources.aertslab.org/cistarget/motif2tf/motifs-v9-nr.hgnc-m0.001-o0.0.tbl. The source data underlying Figs. 2b, 4d, and 4f are provided as a source data file.

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

## Acknowledgements

The authors are grateful to the patients of the senior author (J.W.H.) who generously contributed samples for this research. The authors acknowledge the support of the Biostatistics & Bioinformatics and Oncogenomics Shared Resources at the Sylvester Comprehensive Cancer Center, and the University of Miami Center for Computational Science. The authors thank Dr. Jian-Amadi and Dr. Keene from University of Washington for providing tissue samples and clinical information. This work was supported by National Cancer Institute grant R01 CA125970 (J.W.H.), Research to Prevent Blindness, Inc. Senior Scientific Investigator Award (J.W.H.), Melanoma Research Foundation Established Investigator Award (J.W.H.), the University of Miami Miller School of Medicine Medical Scientist Training Program (M.A.D., D.A.R.), the University of Miami Sheila and David Fuente Graduate Program in Cancer Biology (M.A.D., D.A.R.), the Center for Computational Science Fellowship (M.A.D.), and a generous gift from Dr. Mark J. Daily (J.W.H). The Bascom Palmer Eye Institute received funding from NIH Core Grant P30EY014801 and a Research to Prevent Blindness Unrestricted Grant. The Sylvester Comprehensive Cancer Center also received funding from the National Cancer Institute Core Support Grant P30CA240139.

## Author contributions

M.A.D. analyzed and interpreted the data and wrote the manuscript. D.A.R., S.K., J.N.K., and H.S. analyzed and interpreted the data and edited the manuscript. A.S.L and L.G.F provided metastatic uveal melanoma samples. M.I.S. collected and prepared samples. C.L.D. maintained clinical database and oversaw regulatory compliance. J.W.H. designed and supervised the project, provided clinical samples, interpreted the data and wrote the manuscript.

## Competing interests

J.W.H. is the inventor of intellectual property related to prognostic testing for uveal melanoma. He is a paid consultant for Castle Biosciences, licensee of this intellectual property, and he receives royalties from its commercialization. H.S. is employed by Cell IDx Inc. and owns shares in the company. The other authors declare no competing interests.
