## [Peer Review File · Nature Communications]

Reviewers' comments:

Reviewer #1 (Remarks to the Author):

I think this is an interesting paper that shows the complexity of the tumor-immune microenvironment in Uveal Melanoma. The authors make a compelling case for the dynamic genomic landscape that this disease sets up. The paper is well written and the analysis follows the standard approach to analyze single-cell RNASeq and is comprehensive enough considering the large number of single cells involved. I believe this paper deserves to be published without changes and I only made these suggestion below to improve the manuscript.

- Fig 2A. This figure is too small and it is not easy to read the chr legend and associate that against the CNV features in the heatmaps.
- At the end of the article, the authors speculate about the suitability of LAG3 inhibition as a therapeutic option for high-risk UV. There are some controversies about the role of LAG3 vis a vis other immune cells including NK cells (e.g. Immunol Rev. 2017 Mar; 276(1): 80–96). Given the general interest in LAG3 as a target of immunotherapy, the authors could address this in more depth that is currently done in the manuscript.
- In Data Availability, the authors cite an entry in dbGaP. It will be useful, in addition, to provide the raw data in dbGaP, to also provide at least some of the processed data e.g. in GEO (Gene Expression Omnibus).

Reviewer #2 (Remarks to the Author):

In this manuscript, the authors provide single cell analyses of 8 primary and 3 metastatic uveal melanomas. This approach was never reported in this disease, and results are new and interesting. Bioinformatics analyses thoroughly explore both tumor cells and microenvironment. The manuscript is well written and concise. However, this study includes only a small number of samples, which is appropriate given the cost and work load, but as a consequence most of the claims of the manuscript are very poorly supported by data and statistics. Furthermore, in absence of time series, claims of dynamics and co-evolution do not make much sense.

Major comments

- One can be very surprised to observe not a single hepatocyte or other liver cells in metastatic samples. Please explain. Furthermore, the workflow for metastases is not described in the M&M, please complete.
- p.3 "Individual tumours varied greatly in their composition, with cellular complexity increasing from primary class 1 to metastatic class 2 tumours (Fig. 1d)." No stat is provided to justify this statement. Fig1d contains an arrow for increased complexity, whereas two class 2 samples (61 & 64) with complexity as low as class 1 are misplaced according to the arrow. In absence of stats and given the small number of samples that class 2 UM are really more complex than class 1 is doubtful. The comparison with metastases is not fully relevant as microenvironment is completely different and no comparison between class 1 and class 2 mets can be provided.
- p.4 "Taken together, these findings depict a dynamic genomic landscape in which early competition among multiple CNAs gives rise to one of three principal evolutionary trajectories associated with a signature driver mutation – EIF1AX in class 1A, SF3B1 and other splicing mutations in class 1B, and BAP1 in class 2 tumours – with a background of subclones that have not reached evolutionary fixation." However data and analyses provided here support late rather than early competition, as all tumor cells share the same galpaq and BSA events. Please justify.
- p.6 "At the level of individual samples, cells from class 1 tumours are enriched in states 1-4,14-16 and those from class 2 tumours in states 5-13 (Fig. 3d), consistent with earlier findings that class 2 tumours demonstrate greater cellular heterogeneity than class 1 tumours (Fig. 1d)." As

mentioned earlier, these findings are not supported by stats, and I don't see direct relationship between the analysis of TME heterogeneity and states. Specific tumor cell heterogeneity analysis for the t-SNE (PRAME+ and -) could be provided for such claim. Class 2 samples 61 & 64 have states as rich as other class 2.

- p.6 "These observations confirm the class 1/class 2 partition as a fundamental feature of the global molecular landscape of UM, and they help to resolve a long-standing controversy in the field by favoring a model in which cells of class 2 tumours arise from cells of class 1 tumours, rather than from a distinct population of "class 2" uveal melanocytes." This "long-standing controversy" was already fully and directly clarified in recent publications not cited by the authors (Rodrigues et al CCR, 2019 and Shain et al Nat Genet, 2019).

- p.6-7 "T cells are present in all samples, with CD8+ cytotoxic and CD8+ T effector memory cells most prevalent in class 2 primary tumours and in metastatic samples (Fig. 4c)". Please provide stats, even if this was previously described in numerous studies.

- p.8 "These findings, coupled with the absence of PD-L1 and PD-L2 expression in tumour cells (Supplementary Fig. 5a-b), may explain the ineffectiveness of CTLA and PD1 blockade in metastatic UM". The authors never mentioned that UM is one of the less mutated adult tumor (Furney et al Cancer Discov 2013, Robertson et al Cancer Cell 2017 and others), which is the most likely explanation of the absence of efficiency of anticheckpoints.

- p.8 "These findings suggest a dynamic ecosystem with ongoing immunoediting and adaptation of tumour and immune cells as they co-evolve along trajectories associated with specific sets of genomic aberrations" and title "Single cell analysis of uveal melanoma reveals co-evolution of tumour genome and immune microenvironment". As mentioned early, I don't find any data in this manuscript supporting such statement. The authors discovered a very interesting LAG3+CD8 population. This is an original finding. However, what is supporting a dynamic ecosystem ? What is supporting an immunoediting ? Title and discussion have to be change and to follow the data and not pure speculation.

Minor comments

- CNA and CNV are used, please homogenize

- First line p.8 "T-cell receptors" not B-cell

- p.6 "Transcriptional states are identified that are enriched for cells expressing HLA and other immune genes, consistent with previous work, as well as states associated with melanocyte differentiation and TNF-alpha/NF-kB signaling". This sentence is pretty vague: which HLA ? class I I guess. Which immune genes ? which differentiation and signaling genes ? Please precise.

Reviewer #3 (Remarks to the Author):

The authors took a bioinformatic approach and analyzed 59915 cells through single cell RNA and DNA sequencing on 8 primary and 3 metastatic human uveal melanoma samples. The strengths of the paper are the unique tumor population (uveal melanoma is a rare cancer and deserving of more study), elegant bioinformatic analysis, and well written report. However, the paper suffers from broad reaching and unsubstantiated claims based upon the data and evidence provided.

1. Dimensional reduction analysis showed varying expression of the GEP genes on tumor cells and cells of the TME. The author concluded that the accuracy of the GEP test is partly due to its sampling of a transcriptional cross-section of a complex TME. This is acceptable and convincing.

2. Using inferCNV package, the author analyzed scRNA-seq data and concluded that there is a dynamic genomic landscape in which early competition among multiple CNAs gives rise to one of three principal evolutionary trajectories: EIF1AX in class 1A, SF3B1 in class 1B, and BAP1 in class 2 without evolutionary fixation. The fact that class 1 tumors contain subclones of class 2 CNA and vice versa does not mean the potential of class switch and the lack of evolutionary fixation occurring within tumors. The canonical class 1 and 2 copy number alterations are evolving, and they are known to be not mutually exclusive. For example, it is already known that Chromosome

6p gain can be associated with chromosome 3 loss, although rare. It is even known that when chromosome 6p gain coexists with chromosome 3 loss, survival is better than with monosomy 3 alone. It will take more than what the authors has provided to conclude that there is early competition among multiple CNAs which eventually give rise to three principle evolutionary trajectory.

3. Using SCENIC package to analyze scRNA-seq data, the authors performed analysis to identify cis-regulatory elements and coregulatory modules. The author observed that some cells of class 2 tumors can be found in class 1-associated states, whereas almost no cells of class 1 tumors are found in class 2-associated states. The authors then concluded that their findings help to resolve a long-standing controversy favoring class 2 tumors actually arise from class 1 tumors. This conclusion is derived from comparing trajectory branching based on sample obtained in each patient at a single point of time. It is insufficient to support the conclusions the authors made. It is especially unsure if the presence of class 1 state cells in the class 2 tumors would mean that class 2 tumor cells are derived from class 1 tumor cells.

4. The authors also analyzed branching expression trajectories of each sample using Monocle 2. However, a variety of transcriptional status across tumor cell within one patient can hardly suggest dynamic phenotype plasticity, which is defined as a switching but not co-existence between proliferative phenotype and invasive phenotype in cutaneous melanoma cancer cells.

5. The authors performed hierarchical clustering of the immune cells and revealed robust expression of LAG3 and performed immunofluorescent imaging to validate their findings. They also described one case in which B cell expansion is associated with long term dormancy in metastatic lesions. These findings are interesting but require the analysis of more metastatic samples before claiming that LAG3 is a candidate for immune checkpoint blockade in UM.

Reviewer #4 (Remarks to the Author):

The authors have performed important single cell RNA sequencing and multiplex IHC work to characterize the copy number and immune landscape in uveal melanomas. They show rather convincingly that the genotype/bulk RNA expression of tumors is not a fixed, homogeneous population of tumor and immune subsets. Some of the conclusions, however, are a bit too sweeping or premature given the relatively few samples included in the analysis. My comments, largely in the order of the manuscript text, are as follows:

1) Please include detailed clinical data in the supplemental tables. This is particularly important for the primary tumors. What is the median f/u time from initial diagnosis? Did all Class 2 tumors analyzed metastasize? Were all 3 metastatic tumors taken from the liver? It would be interesting to note any differences between, for example, Class 2 samples that did vs didn't metastasize with mature follow up.

2) The authors claim, "Taken together, these findings depict a dynamic genomic landscape in which early competition among multiple CNAs gives rise to one of three principal evolutionary trajectories associated with a signature driver mutation." However, there are no SF3B1 mutant or Class 1B primary tumors. Please clarify that this report only supports this notion for BAP1 and EIF1AX tumors.

3) Please plot absolute numbers of immune cells so the reader can follow along and better understand the claims made about relative abundance/importance of specific immune subsets.

4) The authors claim "T cells are present in all samples, with CD8+ cytotoxic and CD8+ T effector memory cells most prevalent in class 2 primary tumours and in metastatic samples (Fig. 4c)." It appears invalid to make a broad statement like this when the comparison is pitting 2 primary Class 1 samples against 9 Class 2 primaries/mets.

5) The authors suggest LAG3 is the most frequently expressed marker of exhaustion and a potential therapeutic target, even including it as the last sentence in the abstract. This is an intriguing but incompletely supported statement. Only 3 Class 2 primary samples had any exhausted T cells. The authors should show a bar plot with the raw numbers in addition to percentages as noted in comment #3; percentages may be higher but less therapeutically relevant in a tumor with lower amounts of infiltrate. The authors should be careful to note which LAG3 samples were primary vs metastatic (presumably all primary), because as we are all aware, immune landscapes can vary significantly between primary tumors--largely treated effectively with local therapy--and metastatic tumors that shorten lifespans. Lastly, the IHC suggests the majority of LAG3+ cells are not co-expressing CD8+. Do the authors know which cells these are?

6) The authors state "CD14+ monocytes/macrophages are present in all primary and metastatic samples, with CD68+ macrophages displaying a spectrum from M1- to M2-polarization, the latter being enriched in class 2/BAP1mut samples (Fig. 4b-c and Supplementary Fig. 4b), consistent with previous reports from bulk analysis." - The authors should visually depict or write the statistical assessment that supports the statement that there is an enrichment. Figure 4b does not state this and Figure 4c shows relative percentages of M1/M2 macrophages that appear relatively randomly distributed by Class 1 vs 2 status.

7) The authors state, "Intriguingly, the long latency and low metastatic rate of class 1 UMs may result not simply from an inability to proliferate and disseminate, but from effective immune surveillance, possibly as a result of neoantigens generated by EIF1AX and SF3B1 mutations." - This is an intriguing hypothesis - where is the data from this manuscript that supports the notion that Class 1 tumors have more effective immune surveillance? 1 class 1 metastasis with plasma cell expansion seems a bit too anecdotal to make a statement like this.

8) Minor comment - can the authors clarify why 1 sample that underwent scRNA seq did not undergo VDJ recombination repertoire sequencing?

Reviewer #5 (Remarks to the Author):

The manuscript, where the tumour microenvironment of UM cancer samples is interrogated using, mainly, scRNA-seq, is well written in a concise and clear manner. The scientific community will benefit for this study to be publicly available.

However, I would advise for the following points to be clarified:

The parameters used to QC and discard low quality cells were thoroughly specified in the methods section. However, I would advise that distribution plots of key QC indicators are shown, such as number of UMIs, genes and percent of reads associated with mitochondrial reads. Also, to the best of my knowledge, doublet identification has not been assessed.

The authors used Seurat 2 to perform most parts of the analysis which relies on log normalisation to account for differences in cell library size. Given the fact that recent reports advised the suboptimal nature of this normalisation, have the authors considered using more up to date techniques, such as those using regularised negative binomial regression (implemented in sctransform R package).

I can see that the authors chose tSNE as a non-linear dimensional reduction technique for visualisation. Could you please specify the reasons why you have preferred tSNE over UMAP, which preserves both local and global set structure? Also, the authors didn't show a tSNE plot where cells were grouped by sample of origin; it might be a way to clearly identify sample specific batch effects and also determine if non-neoplastic cells were equally distributed among samples. On that note, it was stated that sample UMM059 was processed using Single Cell 3' Library & Gel Bead Kit v2 while the rest was processed using Single Cell 5'. Has any batch effect correction been applied, will this confounding technical effect be an issue given the fact only one sample was run using the 3' kit.

Although an unsupervised cell clustering approach was computed, obtained cell clusters were classified based on known markers. How well the top differentially expressed genes in each cluster correlate with the markers used for annotation. Also, there is no tSNE plot where cells are coloured by cluster, could the authors provide such a plot?

For the trajectory analysis, are there any reasons not to have used the most updated version of Monocle, which the developers advise to use. Also, the manuscript will benefit by showing expression of key class markers across the cell trajectory.

The authors could complement the existing set of analysis with cell genotyping for known variants using both scRNA and DNA sequencing experiments (see techniques like the one described in software like Vartrix at <https://github.com/10XGenomics/vartrix>).

Lastly, could the authors clarify the criteria to select the genes shown in Fig. 4.b and Supplementary Fig. 4b.

Reviewers' comments:

Reviewer #1 (Remarks to the Author):

I think this is an interesting paper that shows the complexity of the tumor-immune microenvironment in Uveal Melanoma. The authors make a compelling case for the dynamic genomic landscape that this disease sets up. The paper is well written, and the analysis follows the standard approach to analyze single-cell RNASeq and is comprehensive enough considering the large number of single cells involved. I believe this paper deserves to be published without changes and I only made these suggestions below to improve the manuscript.

- Fig 2A. This figure is too small and it is not easy to read the chr legend and associate that against the CNV features in the heatmaps.

Response: Thank you for this suggestion. We have corrected the figure panel and made it more legible.

- At the end of the article, the authors speculate about the suitability of LAG3 inhibition as a therapeutic option for high-risk UV. There are some controversies about the role of LAG3 vis a vis other immune cells including NK cells (e.g. Immunol Rev. 2017 Mar; 276(1): 80–96). Given the general interest in LAG3 as a target of immunotherapy, the authors could address this in more depth that is currently done in the manuscript.

Response: Thank you for this suggestion. We have addressed this article (PMID: 28258692) and discussed *LAG3* in more depth in the discussion (Page 8, Paragraph 2).

- In Data Availability, the authors cite an entry in dbGaP. It will be useful, in addition, to provide the raw data in dbGaP, to also provide at least some of the processed data e.g. in GEO (Gene Expression Omnibus).

Response: Thank you for providing this suggestion. We have additionally deposited the processed data in GEO under accession GSE139829. This has also been added in the manuscript Data Availability section.

Reviewer #2 (Remarks to the Author):

In this manuscript, the authors provide single cell analyses of 8 primary and 3 metastatic uveal melanomas. This approach was never reported in this disease, and results are new and interesting. Bioinformatics analyses thoroughly explore both tumor cells and microenvironment. The manuscript is well written and concise. However, this study includes only a small number of samples, which is appropriate given the cost and work load, but as a consequence most of the claims of the manuscript are very poorly supported by data and statistics. Furthermore, in absence of time series, claims of dynamics and co-evolution do not make much sense.

Response: Since it is not feasible in the clinic to biopsy the same tumor on multiple occasions to create a time series, it is common to infer tumor evolution using methods used here and elsewhere. Nevertheless, we appreciate this criticism and have modified our claims accordingly, as described in detail below.

Major comments

- One can be very surprised to observe not a single hepatocyte or other liver cells in metastatic samples. Please explain.

Response: Hepatocytes and cholangiocytes are preferentially lost during several steps in the single cell preparation process, including cell dissociation and density gradient purification (PMID: 30348985). Since we were not interested in these cell types, we did not perform additional steps needed to preserve these cells for single cell analysis. Further, we intentionally sampled tumor tissue far from the tumor-liver interface in these surgically resected specimens, taking great care to avoid normal liver tissue. We show below a representative photomicrograph of each metastatic sample below, showing how the areas that we sampled (indicated by an asterisk) were comprised of dense populations of tumor cells with few liver cells. We have now clarified this information in the Methods under the section entitled “Patients and sample collection.”

BSSR0022

UMM067L

UMM041L

Furthermore, the workflow for metastases is not described in the M&M, please complete.

Response: This has been added (Methods, “Patients and sample collection” section).

- p.3 “Individual tumours varied greatly in their composition, with cellular complexity increasing from primary class 1 to metastatic class 2 tumours (Fig. 1d).” No stat is provided to justify this statement. Fig1d contains an arrow for increased complexity, whereas two class 2 samples (61 & 64) with complexity as low as class 1 are misplaced according to the arrow. In absence of stats and given the small number of samples that class 2 UM are really more complex than class 1 is doubtful. The comparison with metastases is not fully relevant as microenvironment is completely different and no comparison between class 1 and class 2 mets can be provided.

Response: Since this was a very minor point that was not necessary to support our main claims, we have removed this statement from the text, and we removed the arrow from the figure.

- p.4 “Taken together, these findings depict a dynamic genomic landscape in which early competition among multiple CNAs gives rise to one of three principal evolutionary trajectories associated with a signature driver mutation – EIF1AX in class 1A, SF3B1 and other splicing mutations in class 1B, and BAP1 in class 2 tumours – with a background of subclones that have not reached evolutionary fixation.” However data and analyses provided here support late rather than early competition, as all tumor cells share the same g-alpha-q and BSA events. Please justify.

Response: We apologize for any confusion in this statement, and we now clarify in the text. It was previously shown and published prior to our current work that all canonical driver aberrations are usually present in ~100% of tumor cells at the time of tumor resection, supporting early competition and evolutionary fixation (PMID 28810145, PMID 29317634). Our current data go beyond this finding and reveal ongoing subclonal evolution of non-canonical aberrations that seems to be neutral with respect to the primary tumor but may be relevant to metastatic progression. We have modified the text accordingly (Page 5, 1st paragraph).

- p.6 “At the level of individual samples, cells from class 1 tumours are enriched in states 1-4, 14-16 and those from class 2 tumours in states 5-13 (Fig. 3d), consistent with earlier findings that class 2 tumours demonstrate greater cellular heterogeneity than class 1 tumours (Fig. 1d).” As mentioned earlier, these findings are not supported by stats, and I don’t see direct relationship between the analysis of TME heterogeneity and states. Specific tumor cell heterogeneity analysis for the t-SNE (PRAME+ and -) could be provided for such claim. Class 2 samples 61 & 64 have states as rich as other class 2.

Response: As mentioned in the prior response regarding cellular heterogeneity, we have removed this statement.

- p.6 “These observations confirm the class 1/class 2 partition as a fundamental feature of the global molecular landscape of UM, and they help to resolve a long-standing controversy in the field by favoring a model in which cells of class 2 tumours arise from cells of class 1 tumours, rather than from a distinct population of “class 2” uveal melanocytes.” This “long-standing

controversy” was already fully and directly clarified in recent publications not cited by the authors (Rodrigues et al CCR, 2019 and Shain et al Nat Genet, 2019).

Response: Those studies compared primary with metastatic tumors, and neither investigated gene expression profile classification, so they could not answer the question of whether class 1 primary tumors give rise to class 2 tumors. Nevertheless, since this is a minor point, we prefer simply to remove the statement.

- p.6-7 “T cells are present in all samples, with CD8+ cytotoxic and CD8+ T effector memory cells most prevalent in class 2 primary tumours and in metastatic samples (Fig. 4c)”. Please provide stats, even if this was previously described in numerous studies.

Response: We have now provided statistics as requested (Page 6, 2nd paragraph).

- p.8 “These findings, coupled with the absence of PD-L1 and PD-L2 expression in tumour cells (Supplementary Fig. 5a-b), may explain the ineffectiveness of CTLA and PD1 blockade in metastatic UM”. The authors never mentioned that UM is one of the less mutated adult tumor (Furney et al Cancer Discov 2013, Robertson et al Cancer Cell 2017 and others), which is the most likely explanation of the absence of efficiency of anticheckpoints.

Response: While it is true that most uveal melanomas have a low mutation burden, it is unproven that mutation burden alone explains the lack of efficacy of checkpoint inhibitors in this cancer. Indeed, there is growing evidence that mutation burden alone is not the only important predictor of response to checkpoint inhibitors and can actually anticorrelate with response (PMID 31307554; <https://doi.org/10.1038/s41591-019-0639-4>). Additionally, our novel finding of clonally expanded T cells and/or plasma cells in multiple samples indicates that immune cells are capable of recognizing antigens in uveal melanomas despite low mutation burden. Further, studies such as ours are needed to gain new insights into the TME so that new strategies for immunotherapy can be developed for “cold” tumors such as uveal melanoma. We have now explained this in the text, added the phrase “in part,” and clarify that a key goal of this project was to identify other contributing factors to low immunogenicity that might be amenable to therapeutic manipulation, such as the finding of LAG3 expression (Page 8, 2nd Paragraph).

- p.8 “These findings suggest a dynamic ecosystem with ongoing immunoediting and adaptation of tumour and immune cells as they co-evolve along trajectories associated with specific sets of genomic aberrations” and title “Single cell analysis of uveal melanoma reveals co-evolution of tumour genome and immune microenvironment”. As mentioned early, I don’t find any data in this manuscript supporting such statement. The authors discovered a very interesting LAG3+CD8 population. This is an original finding. However, what is supporting a dynamic ecosystem ? What is supporting an immunoediting ? Title and discussion have to be change and to follow the data and not pure speculation.

Response: We have removed the sentence in question and changed the title to “Single cell sequencing reveals new evolutionary complexity in uveal melanoma.”

Minor comments

- CNA and CNV are used, please homogenize

Response: We have changed all to CNV.

- First line p.8 "T-cell receptors" not B-cell

Response: We have corrected this.

- p.6 "Transcriptional states are identified that are enriched for cells expressing HLA and other immune genes, consistent with previous work, as well as states associated with melanocyte differentiation and TNF-alpha/NF-kB signaling". This sentence is pretty vague: which HLA ? class I I guess. Which immune genes ? which differentiation and signaling genes ? Please precise.

Response: We have specified HLA class I as well as the specific immune genes (Page 6, 1st paragraph).

Reviewer #3 (Remarks to the Author):

The authors took a bioinformatic approach and analyzed 59915 cells through single cell RNA and DNA sequencing on 8 primary and 3 metastatic human uveal melanoma samples. The strengths of the paper are the unique tumor population (uveal melanoma is a rare cancer and deserving of more study), elegant bioinformatic analysis, and well written report. However, the paper suffers from broad reaching and unsubstantiated claims based upon the data and evidence provided.

Response: We thank the reviewer for positive comments, and we have modified our claims in accordance with the reviewer's suggestions, as described below.

1. Dimensional reduction analysis showed varying expression of the GEP genes on tumor cells and cells of the TME. The author concluded that the accuracy of the GEP test is partly due to its sampling of a transcriptional cross-section of a complex TME. This is acceptable and convincing.

Response: Thank you for your comment.

2. Using inferCNV package, the author analyzed scRNA-seq data and concluded that there is a dynamic genomic landscape in which early competition among multiple CNAs gives rise to one of three principal evolutionary trajectories: EIF1AX in class 1A, SF3B1 in class 1B, and BAP1 in class 2 without evolutionary fixation. The fact that class 1 tumors contain subclones of class 2 CNA and vice versa does not mean the potential of class switch and the lack of evolutionary fixation occurring within tumors. The canonical class 1 and 2 copy number alterations are evolving, and they are known to be not mutually exclusive. For example, it is already known that Chromosome 6p gain can be associated with chromosome 3 loss, although rare. It is even

known that when chromosome 6p gain coexists with chromosome 3 loss, survival is better than with monosomy 3 alone. It will take more than what the authors has provided to conclude that there is early competition among multiple CNAs which eventually give rise to three principle evolutionary trajectory.

Response: Please see response to a similar comment from Reviewer 2. In summary, the statement in question was shown in other publications. We were simply stating this information to put it into context with the new findings of this manuscript. We have now clarified this point (Page 5, 1st paragraph).

3. Using SCENIC package to analyze scRNA-seq data, the authors performed analysis to identify cis-regulatory elements and coregulatory modules. The author observed that some cells of class 2 tumors can be found in class 1-associated states, whereas almost no cells of class 1 tumors are found in class 2-associated states. The authors then concluded that their findings help to resolve a long-standing controversy favoring class 2 tumors actually arise from class 1 tumors. This conclusion is derived from comparing trajectory branching based on sample obtained in each patient at a single point of time. It is insufficient to support the conclusions the authors made. It is especially unsure if the presence of class 1 state cells in the class 2 tumors would mean that class 2 tumor cells are derived from class 1 tumor cells.

Response: Since this is a minor point, we prefer simply to remove the statement.

4. The authors also analyzed branching expression trajectories of each sample using Monocle 2. However, a variety of transcriptional status across tumor cell within one patient can hardly suggest dynamic phenotype plasticity, which is defined as a switching but not co-existence between proliferative phenotype and invasive phenotype in cutaneous melanoma cancer cells.

Response: We agree that the word “dynamic” may not be substantiated and have removed it. The definition of phenotypic plasticity offered by the reviewer is based on *in vitro* experiments prior to the availability of single-cell sequencing technology. We believe that our data provide evidence for potential phenotypic plasticity *in vivo* and prefer to mention this, but we have modified the statement in accordance with this comment (Page 6, 1st paragraph).

5. The authors performed hierarchical clustering of the immune cells and revealed robust expression of LAG3 and performed immunofluorescent imaging to validate their findings. They also described one case in which B cell expansion is associated with long term dormancy in metastatic lesions. These findings are interesting but require the analysis of more metastatic samples before claiming that LAG3 is a candidate for immune checkpoint blockade in UM.

Response: We believe that the finding of *LAG3* expression in both primary and metastatic samples support this claim, especially since checkpoint inhibitors are now being investigated in clinical trials of high risk UM in the adjuvant setting (NCT03528408). We show that *LAG3* is expressed in 100% of 22 tumors analyzed by single-cell analysis, IHC, or both. This is a substantial number of cases for this rare tumor type, and we believe that this supports our claim. To address the reviewer’s comment, however, we have changed the wording to “a potential candidate for immune checkpoint blockade in UM.”

Reviewer #4 (Remarks to the Author):

The authors have performed important single cell RNA sequencing and multiplex IHC work to characterize the copy number and immune landscape in uveal melanomas. They show rather convincingly that the genotype/bulk RNA expression of tumors is not a fixed, homogeneous population of tumor and immune subsets. Some of the conclusions, however, are a bit too sweeping or premature given the relatively few samples included in the analysis. My comments, largely in the order of the manuscript text, are as follows:

1) Please include detailed clinical data in the supplemental tables. This is particularly important for the primary tumors. What is the median f/u time from initial diagnosis? Did all Class 2 tumors analyzed metastasize? Were all 3 metastatic tumors taken from the liver? It would be interesting to note any differences between, for example, Class 2 samples that did vs didn't metastasize with mature follow up.

Response: These data have now been added to Supplementary Table 1. Follow-up is short, since the samples were collected prospectively (median follow up of primary tumor patients, 5 months). Two of the class 2 primary tumors have metastasized. All 3 metastatic tumor samples were taken from the liver. We agree that any differences will be interesting to analyze once the data are more mature.

2) The authors claim, "Taken together, these findings depict a dynamic genomic landscape in which early competition among multiple CNAs gives rise to one of three principal evolutionary trajectories associated with a signature driver mutation." However, there are no SF3B1 mutant or Class 1B primary tumors. Please clarify that this report only supports this notion for BAP1 and EIF1AX tumors.

Response: We have corrected this in the manuscript.

3) Please plot absolute numbers of immune cells so the reader can follow along and better understand the claims made about relative abundance/importance of specific immune subsets.

Response: Thank you for providing this suggestion. We have provided a new Supplementary Figure 8 with the absolute numbers of immune cells plotted.

4) The authors claim "T cells are present in all samples, with CD8+ cytotoxic and CD8+ T effector memory cells most prevalent in class 2 primary tumours and in metastatic samples (Fig. 4c)." It appears invalid to make a broad statement like this when the comparison is pitting 2 primary Class 1 samples against 9 Class 2 primaries/mets.

Response: The reviewer's point is well taken. We removed this claim and now simply provide summary statistics (Page 6, 2nd paragraph).

5) The authors suggest LAG3 is the most frequently expressed marker of exhaustion and a potential therapeutic target, even including it as the last sentence in the abstract. This is an intriguing but incompletely supported statement. Only 3 Class 2 primary samples had any exhausted T cells. The authors should show a bar plot with the raw numbers in addition to

percentages as noted in comment #3; percentages may be higher but less therapeutically relevant in a tumor with lower amounts of infiltrate. The authors should be careful to note which LAG-3 samples were primary vs metastatic (presumably all primary), because as we are all aware, immune landscapes can vary significantly between primary tumors--largely treated effectively with local therapy--and metastatic tumors that shorten lifespans.

Response: We apologize for any confusion and wish to clarify. All 18 samples analyzed by IHC (two of which were metastatic samples that are now indicated in the figure legend) contained LAG3+/CD8+ T cells. Further, LAG3 was the most prevalent exhaustion marker in all of these samples (Fig. 4f). There were 3 class 2 samples that contained clonally expanded T cells by V(D)J analysis (Fig. 4d), which is an independent finding from our claim regarding LAG3. We have clarified this in the text (Page 8, 2nd paragraph). We have also provided the raw numbers for the bar plot as requested and also mentioned in comment #3.

Lastly, the IHC suggests the majority of LAG3+ cells are not co-expressing CD8+. Do the authors know which cells these are?

Response: Yes. As shown in the tSNE plot of the scRNA-seq data below, LAG3 expression is present in some CD4+ T cells, FOXP3+ regulatory T cells, NK cells, and macrophage/monocytes. We have added this to the text (Page 7, 1st paragraph) and added this to Supplementary Figure 10.

6) The authors state "CD14+ monocytes/macrophages are present in all primary and metastatic samples, with CD68+ macrophages displaying a spectrum from M1- to M2-polarization, the latter being enriched in class 2/BAP1mut samples (Fig. 4b-c and Supplementary Fig. 4b), consistent with previous reports from bulk analysis." - The authors should visually depict or write the statistical assessment that supports the statement that there is an enrichment. Figure 4b does not state this and Figure 4c shows relative percentages of M1/M2 macrophages that appear relatively randomly distributed by Class 1 vs 2 status.

Response: The reviewer's point is well taken, and we have removed the claim of enrichment, which is not relevant to our main points.

7) The authors state, "Intriguingly, the long latency and low metastatic rate of class 1 UMs may result not simply from an inability to proliferate and disseminate, but from effective immune surveillance, possibly as a result of neoantigens generated by EIF1AX and SF3B1 mutations." - This is an intriguing hypothesis – where is the data from this manuscript that supports the notion that Class 1 tumors have more effective immune surveillance? 1 class 1 metastasis with plasma cell expansion seems a bit too anecdotal to make a statement like this.

Response: We agree that the observation is somewhat anecdotal, but it is a very provocative and potentially valuable insight that we believe deserves comment. We have re-worded this statement to clearly indicate that it is only a speculation that will require further validation.

8) Minor comment - can the authors clarify why 1 sample that underwent scRNA seq did not undergo VDJ recombination repertoire sequencing?

Response: That was the first sample to be analyzed, and it was run using the single cell 3' gene expression, which is not compatible with the single cell VDJ sequencing kit. The 5' gene expression kit was not available at that time. The rest of the samples were run using the single cell 5' gene expression kit, which is compatible with VDJ recombination repertoire sequencing. This information was mentioned in the methods but has been further clarified (Methods, Single-cell RNA sequencing).

Reviewer #5 (Remarks to the Author):

The manuscript, where the tumour microenvironment of UM cancer samples is interrogated using, mainly, scRNA-seq, is well written in a concise and clear manner. The scientific community will benefit for this study to be publicly available.

However, I would advice for the following points to be clarified:

The parameters used to QC and discard low quality cells were thoroughly specified in the methods section. However, I would advice that distribution plots of key QC indicators are shown, such as number of UMIs, genes and percent of reads associated with mitochondrial reads. Also, to the best of my knowledge, doublet identification has not been assessed.

Response: As requested, we have added the QC plots for number of UMIs, number of genes, and percent mitochondrial content in Supplementary Figure 1. Additionally, we assessed doublets using the DoubletFinder algorithm recently published in Cell Systems (PMID: 30954475). We also provide a plot of the doublets in in Supplementary Figure 1. We observed very few doublets in our dataset except for the macrophage/monocyte population which are known to be sticky. The doublets observed have no effect on the conclusions of the manuscript.

The authors used Seurat 2 to perform most parts of the analysis which relies on log normalisation to account for differences in cell library size. Given the fact that recent reports advised the suboptimal nature of this normalisation, have the authors considered using more up to date techniques, such as those using regularised negative binomial regression (implemented in sctransform R package).

Response: We have used sctransform in some of our other datasets, but it was not available at the time our manuscript was submitted, and based on our experience, we do not believe that it would result in substantial differences in clustering or cell type identification in our tumor type. Further, we are not convinced that the optimal parameters have yet been established for this package, as the authors of this code continue to provide new guidelines. Since a re-analysis at this point would require a complete reworking of the entire manuscript, and since such an analysis is unlikely to yield a meaningful difference in outcome, we prefer to stick with Seurat 2, which has a strong track record in many high impact publications.

I can see that the authors chose tSNE as a non-linear dimensional reduction technique for visualisation. Could you please specify the reasons why you have preferred tSNE over UMAP, which preserves both local and global set structure?

Response: We routinely use both tSNE and UMAP to visualize our data, but in this case we believed that tSNE provided a better and more intuitive visualization of the data. Since (1) these are both simply visualization techniques, rather than analytical methods, (2) both methods are accepted by the research community, (3) tSNE has a strong track record in high impact publications, and (4) UMAP did not change our interpretations or conclusions, we prefer to present the tSNE plots. However, we can include the UMAP plots as a supplementary file if requested.

Also, the authors didn't not show a tSNE plot were cells were group by sample of origin; it might be a way to clearly identify sample specific batch effects and also determine if non-neoplastic cells were equally distributed among samples. On that note, it was stated that sample UMM059 was processed using Single Cell 3' Library & Gel Bead Kit v2 while the rest was processed using Single Cell 5'. Has any batch effect correction been applied, will this confounding technical effect it be an issue given the fact only one sample was run using the 3' kit.

As requested, we have added a tSNE plot with the cells grouped by the sample of origin (Supplementary Figure 2a). Since minimal batch effect was observed with regards to UMM059,

we chose not to apply a batch correction, and this did not affect any conclusions in the manuscript.

Although an unsupervised cell clustering approach was computed, obtained cell clusters were classified based on known markers. How well the top differentially expressed genes in each cluster correlate with the markers used for annotation. Also, there is no tSNE plots where cells are coloured by cluster, could the authors provide such a plot?

Response: We now provide the top differentially expressed gene lists as Supplementary Tables 3 and 6 and they correlate well with the markers used for annotation. We also provide a tSNE plot colored by the clusters determined by unsupervised clustering (Supplementary Figure 2b).

For the trajectory analysis, are there any reasons not to have used the most updated version of Monocle, which the developers advice to use. Also, the manuscript will benefit by showing expression of key class markers across the cell trajectory.

Response: At the time the manuscript was submitted, Monocle 2 was the latest version of the algorithm, and Monocle 3 was only available in alpha version, which was not very good and was subsequently recalled. The beta version of Monocle 3 was released more recently, but we and others in the github community encountered issues implementing the code with respect to its package dependencies. The final version of Monocle 3 only became available in the past month, and it has not been completely vetted. Given that (1) we still have concerns about what has become a “moving target” with regard to Monocle 3, (2) Monocle 2 has a proven track record in many high profile publications, and (3) it is highly unlikely that Monocle 3 would improve or substantially change our conclusions, we prefer to stick with Monocle 2.

As requested, we now provide plots showing the expression of key markers along the cell trajectory (Supplementary Figure 5). As expected, the expression of key class markers are identified on the trajectory according to GEP class.

The authors could complement the existing set of analysis with cell genotyping for known variants using both scRNA and DNA sequencing experiments (see techniques like the one described in software like Vartrix at <https://github.com/10XGenomics/vartrix>).

Response: We have performed VarTrix analysis using our scRNA data looking at key driver mutations, including *GNAQ*, *GNA11*, *BAP1*, *SF3B1*, and *EIF1AX*. Unfortunately, most cells did not have good coverage at these mutation sites, since the 10X technology generates short reads from the 5' or 3' end of the cDNA template. However, one of our samples (UMM065), which contained a mutation in *EIF1AX* rendered at least one read at the mutated site in 2,506 cells (see figure below). All other samples yielded between 0 to 46 cells with at least one read of a given mutation. While interesting, we do not believe that this enhances the manuscript, given the limited nature of this finding. To overcome the current limitations of this type of analysis, we are working with 10X to optimize the application of scDNA for mutation calling, but this is still a work in progress and will be the subject of another manuscript.

VarTrix Analysis of EIF1AX Mutation in UMM065

Lastly, could the authors clarify the criteria to select the genes shown in Fig. 4.b and Supplementary Fig. 4b.

Response: The genes selected in these figures were from the differentially expressed markers list, literature review, and our knowledge of important immune markers. Additionally we cross-referenced this information with that provided from the FindAllMarkers() Seurat differential expression analysis using the default two-sided non-parametric Wilcoxon rank sum test. We now provide clarification on this in the Methods section (Methods, “Single-cell RNA sequencing analysis” section).

REVIEWERS' COMMENTS:

Reviewer #1 (Remarks to the Author):

The authors have satisfactorily addressed my comments in the original review. I don't have any additional comments and I think the paper is ready for publication.

Reviewer #2 (Remarks to the Author):

In this revised version, the authors addressed most of the concerns by removing sentences and paragraphs, rather than realizing more experiments and analyses to defend their claims, leading to a very descriptive work.

Major comments

My previous review was: p.8 "These findings suggest a dynamic ecosystem with ongoing immunoediting and adaptation of tumour and immune cells as they co-evolve along trajectories associated with specific sets of genomic aberrations" and title "Single cell analysis of uveal melanoma reveals co-evolution of tumour genome and immune microenvironment". As mentioned early, I don't find any data in this manuscript supporting such statement. The authors discovered a very interesting LAG3+CD8 population. This is an original finding. However, what is supporting a dynamic ecosystem ? What is supporting an immunoediting ? Title and discussion have to be change and to follow the data and not pure speculation. In response the authors changed the title, but did not remove or argue for the first sentence of their discussion "These findings suggest a dynamic ecosystem with ongoing immunoediting and adaptation of tumour and immune cells as they co-evolve along trajectories associated with specific sets of genomic aberrations".

Transcriptional Trajectory Analysis

- Actually the analysis by Scenic is another way to distinguish class 1 from class 2, and has nothing to do with plasticity (Fig3a).
- Looking at Fig3C, a patient bias is relatively clear. No details are provided about combining the 11 samples, except it was done on Seurat 2.3.4. Why this version, while Seurat 3 was used for analyzing CD8 subset ? Seurat 3 improved a lot merging samples and should be used (Stuart, Butler, et al., Cell 2019; 10.1016/j.cell.2019.05.031).

Reviewer #3 (Remarks to the Author):

The authors have satisfactorily addressed by comments. This is a thoughtful and in depth analysis of uveal melanoma tumors.

Reviewer #4 (Remarks to the Author):

The authors have done a good job addressing my concerns, and in my opinion, the concerns raised by fellow reviewers. This is an overall very important hypothesis-generating manuscript that should advance the field of UM biology and therapeutics.

My only minor comment is that the authors should more clearly highlight the fact that 18 additional, distinct samples underwent IHC staining for LAG3/etc in Figure 4 legend and in the manuscript text so readers can follow the total sample set.

Thank you for eliciting my opinions.
Alexander Shoushtari

Reviewer #5 (Remarks to the Author):

From my point of view, all points raised in the previous round of review have been satisfactorily addressed.

I would recommend for the manuscript to be accepted for publication in the journal.